# Taste and Health Information on Fast Food Menus to Encourage Young Adults to Choose Healthy Food Products: An Experimental Study

**DOI:** 10.3390/ijerph17197139

**Published:** 2020-09-29

**Authors:** Frans Folkvord, Maud van der Zanden, Sara Pabian

**Affiliations:** 1Open Evidence Research, Barcelona, 08018 Barcelona, Spain; 2Tilburg School of Humanities and Digital Sciences, Communication and Cognition, Tilburg University, 5037 AB Tilburg, The Netherlands; maudvanderzanden@hotmail.com (M.v.d.Z.); sara.pabian@uantwerpen.be (S.P.); 3Department of Communication Sciences, Research Group MIOS (Media, ICT, and Interpersonal Relations in Organisations and Society), Faculty of Social Sciences, 2000 Antwerp, Belgium

**Keywords:** young adults, healthy food choice, menu information, fast food restaurants

## Abstract

Currently, a great number of people have an unhealthy dietary intake, leading to chronic diseases. Despite the high prevalence of obesity and people being overweight, only a few strategies to promote healthier food products have been proven effective. Therefore, the objective of this study was to test the effect of the presence of health information and its integration into a fast food menu context on young adults’ healthy food choices. An experimental between-subjects design consisting of three conditions—subtle, explicit, and no health information—was conducted among 142 participants aged 18 to 24 (*M*_age_: 21.49, *SD* = 1.77). The results showed that when health information about healthy products was provided, the level of integration of the information into the menu context had an effect. More specifically, participants exposed to explicit health information about healthy products provided on the fast food menu were more likely to choose a healthy food product compared to participants exposed to subtle integrated health information. No interaction effect for moderating factors was found. In line with the healthy food promotion model, the findings suggest that the provision of explicit health information on healthy products stimulates healthy food choices in a fast food environment.

## 1. Introduction

Currently, worldwide many young adults are overweight or obese [1,2]. During the past three decades, obesity among young adults has risen to an extent that it has become one of the biggest public health concerns worldwide [1,2,3,4], although young adults themselves believe they should eat healthier and know why they should do so [5]. However, eating healthily is not that easy since young adults are targeted by many advertisements that are designed to encourage them to consume energy-dense foods [6,7,8,9,10,11]. In addition, most young adults have little knowledge of the nutrients they consume in daily life and therefore find it hard to distinguish healthy from unhealthy food products [12,13,14]. Consequently, they develop eating habits that differ from patterns that are recommended for meeting national dietary reference intakes [15,16,17].

Considering the societal, health, and economic consequences of the current food system [1], the food industry must take responsibility for young adults’ health by focusing on the promotion of healthy food products they are selling rather than promoting unhealthy food products [11]. Since eating fast food is a popular lifestyle activity among young adults due to the low prices and convenience they offer, and they are considered to play a major role in the obesity epidemic [6,7,8,9,18,19,20]. In addition to the ethical considerations of a healthier society, considering that fast food restaurants also sell low-calorie and healthier foods, such as salad, they will benefit from promoting healthier foods as well. Hence, fast food restaurants have the possibility to steer young adults toward the direction of healthier products, which can be done effectively while keeping unhealthy (food) products on the menu [21].

### 1.1. Theoretical Background

The healthy food promotion model [8] describes that by increasing attention toward healthier foods and improving the reinforcing value (e.g., liking and wanting) through food promotion, people will be more likely to buy and consume these foods. As a consequence of this increased consumption, a reciprocal relation between food promotion with eating behavior will occur [6], which in time will potentially lead to a normalization of the intake of healthier foods, whereby habit formation of a healthier diet is created. A key challenge is to promote healthier food without generating a negative taste perception that is often associated with healthy products [8].

The perception that healthy food is less tasty because it tastes natural or light is based on earlier experiences with this food [22,23,24]. It is demonstrated that the provision of taste information can have a positive effect on the perceived tastiness of healthy food products and the selection of these products, since the perceived tastiness of food products is one of the most important determinants of food choices [12,25]. Furthermore, when consumers evaluate menu products and form purchase decisions, they are both sensitive to and willing to use health information when this is directly available [26]. More specifically, some studies have shown that health information somehow increases the number of consumers choosing healthy/healthier food products [26,27,28], while other studies have shown that the information can cause consumers to compensate calories with other menu products (e.g., side orders) or products at another moment of the day (e.g., late-night snacks), thereby reducing the positive effects of healthy food consumption [19]. Consumers that choose (some) healthier food products based on health information might even have the tendency to consume more calories than they would do in the first place [6].

Whether the provision of health information is effective might depend on the level of integration of the information into the menu context and the level of health consciousness of consumers. More specifically, a recent systematic review showed that priming salience information about the healthiness of food increased healthier food choices [29], suggesting that health information is effective in encouraging consumers to choose a healthy snack product when the information is provided in an explicit manner, as compared to being subtle. However, some studies have shown that subtly labeling a food as healthy works more effectively as opposed to explicitly labeling [30], probably due to differences in the extent to which consumers experience it as a threat to their freedom of choice [30,31,32].

It is expected that health information is more likely to be experienced as a threat among young adults with low levels of health consciousness who consider health as insignificant and want to maintain their hedonic lifestyle, than among young adults with high levels of health consciousness; the latter are satisfied with any health information provided [13,33,34]. Health consciousness is a personality trait that is characterized by being concerned about personal health [35,36]. Consumers with this personality trait are interested in healthy food products and pay a lot of attention to their health [37,38], have positive attitudes toward wellness activities, are willing to live a wellness-oriented lifestyle, and are actively thinking about their food intake and physical condition [36,37,38].

According to the young people’s processing of commercial media content framework (PCMC [39]), the level of integration of health information into the menu context could influence the effectiveness of the information among young adults due to a difference in how the information is processed. More specifically, the level of integration of information in its context affects young adults’ motivation, ability, and opportunity to process information [39]. This, in turn, influences whether young adults are able to pay attention to the information and become aware of the persuasiveness of the information, and influences whether resistance will occur [40,41,42,43]. Thus, the level of integration may be able to explain the effectiveness of health information on healthy food choice when focusing on the ways in which young adults process health information that is integrated into the menu context at a certain level. More specifically, when information is not integrated into a certain context (explicit information), it is easier for consumers to pay attention to and be aware of the information; and to become motivated and able to process the information [43]. When adults are exposed to explicit information, attitude and/or behavior change is likely to occur due to the cognitive processing of the information, when it is aligned with their intrinsic values and norms [28,31,44,45,46]. Providing consumers with effective information on food decisions might be relevant for fast food restaurants that are willing to encourage young adults to choose a healthy product. The current study aims to provide a state-of-the-art tool to encourage consumers to buy healthier food in fast-food restaurants.

### 1.2. Current Study

The current study examines experimentally the effects of health information of healthy products on healthy food choice to find out whether fast food restaurants should provide the information on their menus when taste information is already provided. Restaurants that provide information on their menu might be able to help young adults while simultaneously accomplishing their own financial goals, as it is a low-cost strategy [47,48]. Based on the literature overview, we hypothesize that participants who are exposed to a menu with health information about the healthy products are more likely to choose a healthy food product than participants who are exposed to a menu without health information of the healthy products. More specifically, we hypothesize that participants exposed to a menu with explicit health information of healthy products are more likely to choose a healthy food product than participants who are exposed to a menu with subtle health information of healthy products.

## 2. Materials and Methods

### 2.1. Experimental Design and Stimulus Material

A between-subjects experimental design was conducted with three different conditions (subtle health information [see Figure 1] vs. explicit health information [see Figure 2] vs. no health information) in an online questionnaire, in which the health information provided for healthy food products on a fast food restaurant’s menu was manipulated. No health information was provided in any of the conditions for the unhealthy food products.

In all three conditions, participants were exposed to a fast food menu with four menu products. All menus contained information that would normally appear on an actual menu (i.e., pictures, names, allergy information, and prices per menu product). The three menus were identical, except for the information that was provided on the right side of two healthy products.

In the condition where subtle health information was provided, participants were exposed to health information similar to the information used in the study conducted by Wagner et al. ([30]; i.e., a red heart with a white checkmark). In the condition where explicit health information was provided, participants saw health information that was perceived as explicit information in pretest 3 (i.e., “healthy X kcal’) and an icon of an arrow pointing toward the product). In all conditions, taste information was provided as “tasty”, based on the study conducted by Turnwald and Crum [48]. While multiple studies provided an example of taste information, not every example could be used for all healthy food products in the current study (e.g., succulent). Thus, the most obvious one-word example was used (i.e., tasty).

The four food products that were shown on the menus in the experiment were derived from the menu of one of the most famous fast food restaurants in the world. This restaurant was chosen for three reasons. Firstly, the restaurant currently provided its customers with nutrient information of all menu products on its corporate website. Secondly, the restaurant offers lower-calorie products that can be ordered for dinner (e.g., 138 [Grilled Chicken Salad] compared to 980 calories [Double Big Tasty]). All products that were not offered as a snack or side product were considered a dinner product. Finally, the restaurant enables its customers to order food via a self-service system. The interface of this system was similar to that of a drive-in and home-delivery system, as customers could order food without being directly in contact with an employee of the restaurant. Using a system similar in the current study provided the researchers the opportunity to simulate the environment in which fast food was being ordered for consumption at (drive-in) restaurants and home.

### 2.2. Participants

In total, 225 Dutch-speaking people participated in the study. Of these people, 40 people did not complete the questionnaire, 34 participants were allergic/intolerant to food or a food substance, and 47 participants were following a diet while data were being collected. As a result, the analytical sample consisted of 142 young adults (The same results were found with the complete sample including these participants who were allergic to food or were on a diet). The sample ranged in age from 18 to 24 years old (*M* = 21.49, *SD* = 1.77) because we were mainly interested in the effects on young adults and consisted of 44 males (31%) from the Netherlands. Participants in this study were randomly obtained via convenience sampling on social media, such as Instagram and Facebook, the social media platforms young adults were using the most during the data collection [49]. Dutch-speaking participants were invited to participate if they were between 18 and 24 years old. By clicking on the link in the invitation, participants accessed the online questionnaire. The participants could fill in and complete this questionnaire at any time between 6 November, 2019, and 11 November, 2019. An *a priori* G*power analysis suggested to have at least 116 participants (effect size = 0.15, alpha = 0.05, power = 0.80, and five predictors).

### 2.3. Procedures

The final experiment was integrated in a Dutch online questionnaire designed in Qualtrics. Participants were randomly assigned by the randomization option in Qualtrics to one of the three conditions (subtle *n =* 42, explicit *n =* 49, or no health information *n =* 51 about the healthy product), whereby active consent was collected and socio-demographic information was assessed. Because both the researcher and the participants were not aware of the allocations of the conditions, this study was double-blinded. Next, participants had to play a simple rhyme game as a distracting task to cover the goal of the experimental study, in which they had to indicate for 11 words which word rhymed with those words. To make sure every participant was able to give the right answers to the questions without much cognitive effort, the game was based on rhyme words that were originally used for children from the upper classes of elementary school [50]. Participants were told that they could take any time they needed and participants had to select one answer from a defined list of choices that was provided for each question (multiple-choice). After one example rhyme question and five official rhyme questions, participants were asked which food product they would choose while being exposed to a menu of the fast food brand, which was the primary outcome of the study. No secondary outcomes were assessed. After choosing one product, participants continued with the other five official rhyme questions.

In the third and last part, participants were asked about their attitude toward the fast food brand, familiarity with the products of the fast food brand, level of health consciousness, allergies and intolerances, diets, and gender. When all questions were answered, participants were thanked for their participation and the real purpose of the study was revealed. The committee for ethical concerns in the Faculty of Humanities and Digital Sciences at the Tilburg University approved the current study.

### 2.4. Pretest

Before actual data were collected, the stimulus materials were tested among 15 participants to find out which food products of the menu should represent the healthy and the unhealthy food products on the menus in the experiment. For 10 menu products, participants had to indicate the perceived healthiness. The two menu products of which the healthiness was perceived the lowest were used in the experiment as the unhealthy products. The two menu products of which the healthiness was perceived the highest were used in the experiment as the healthy products. Therefore, the Grilled Chicken Salad (with an average of 3.50 [*SD* = 0.91], indicating “healthy’’) and the Veggie Falafel Salad (with an average of 3.36 [*SD* = 0.89], indicating “not healthy/not unhealthy’’) represented the healthy products, and the Double Big Tasty (with an average of 1.07 [*SD* = 0.26], indicating “very unhealthy”) and the Meastro Angus^TM^ (with an average of 1.21 [*SD* = 0.56], indicating “very unhealthy”) represented the unhealthy products.

### 2.5. Measures

*Dependent variable*: To measure healthy food choice, participants were asked that if they would visit this fast food restaurant, which food product they would select from the menu. There were two healthy and two unhealthy food products, which were represented by four food products in total. It was recorded for each participant if he/she selected a healthy food product or not, which was the primary outcome measure of the study.

*Health consciousness*. Participants’ health consciousness was measured with 11 items (e.g., “I’m very self-conscious about my health”) on a 7-point Likert scale (1 = *strongly disagree* to 7 = *strongly agree*), on which a higher score indicated a higher level of health consciousness [38]. In the present sample, the reliability of the scale was high (*α* = 0.75).

*Brand attitude*. Brand attitude toward fast food brand was measured with one question (i.e., “Please describe your overall feelings toward the brand, the brand of which the menu was presented earlier in this questionnaire”) using five bipolar affective and evaluative dimensions with a 7-point scale (*unappealing* to *appealing*, *bad* to *good*, *unpleasant* to *pleasant*, *unfavorable* to *favorable*, and *unlikable* to *likable*). In the present sample, the reliability of this multidimensional scale was high (*α* = 0.81). The measure of this attitudinal construct was developed by Spears and Singh [51].

*Familiarity with the products*. Furthermore, familiarity with the products that the fast food brand offers was assessed. Participants’ familiarity with the products was measured with one item (i.e., “How familiar are you with the products this restaurant offers?”) on a 5-point Likert scale (1 = *not at all familiar* to 5 = *extremely familiar*; [52]).

*Frequency of visiting the fast food restaurant*. Final, frequency of visiting the fast food restaurant to buy/eat something was assessed. To measure this, participants were asked how often they visited the restaurant (i.e., “How many times a week/month/year do you visit this fast food restaurant to eat something and/or to buy food?’’, with the response options: 1 = *never*, 2 = *less than annually*, 3 = *once a year*, 4 = *multiple times a year*, 5 = *once a month*, 6 = *multiple times a month*, 7 = *once a week*, 8 = *multiple times a week*, 9 = *every day*, 10 = *I don’t know*; [53]).

### 2.6. Statistical Analysis

Firstly, the assumptions of normality (one-sample Kolmogorov–Smirnov test) and homogeneity of variance (Levene’s test) were tested. In addition, there had to be a linear relationship between the continuous independent variable(s) and logit transformation (log odds) of healthy food choice. To assess whether there was a linear relationship, the Box–Tidwell test was performed. Outlying scores on healthy food choice were estimated that could affect the results by computing residual scores and by testing these scores for Mahal’s distance, Cook’s distance, and leverage scores.

Secondly, a randomization check was performed with a multivariate analysis of variance (MANOVA), in which differences in the distribution between the health information conditions were examined to see if manipulation was successful. In addition, correlations were examined between age, attitude toward fast food brand, familiarity with the products of the fast food brand, health consciousness, gender, and participants’ healthy food choices (bivariate). A relation was considered significant when the *p*-value was < 0.05.

Finally, binomial logistic regression analyses with Wald tests were performed to test the hypotheses in which the effect of health information (condition) on healthy food choice was examined. Significance was suspected when the *p*-value was < 0.05.

## 3. Results

On average, the participants had a somewhat positive attitude toward the fast food brand (*M* = 4.52, *SD* = 1.01). Half of the sample was visiting the fast food brand to eat and/or buy food multiple times a year (49%), followed by once a month (27%), multiple times a month (14%), less than annually (5%), once a week (3%), and never (1%). One participant (1%) did not give an indication of his/her visit frequency. The participants in the sample were on average moderately familiar with the products the fast food brand offered their customers (*M* = 3.82, *SD* = 0.71). Lastly, a mean score of 5.12 on the health consciousness scale indicated that the participants in the sample were on average somewhat health-conscious (*SD* = 0.66).

The results of the one-sample Kolmogorov–Smirnov test showed that the data were not normally distributed (Condition: *D*(142) = 0.24, *p* < 0.001; Health Consciousness: *D*(142) = 0.11, *p* < 0.001; Gender: *D*(142) = 0.44, *p* < 0.001). The assumption of homogeneity of variances was not met either, since the Levene’s test was significant for gender, *F*(2, 139) = 6.20, *p* = 0.003. Therefore, the results of the binomial logistic regression analyses should be interpreted with caution.

On the other hand, the results of the Box–Tidwell test showed that the relationship between health consciousness as a continuous predictor and its log odds was not significant (*p* = 0.869). The Mahal’s distance, Cook’s distance, and leverage scores did not suggest any outlying scores.

A MANOVA test showed (see Table 1) that there were no significant differences between the conditions on age, attitude toward the fast food brand, familiarity with the products of the fast food brand, health consciousness, and gender (*p* > 0.05). According to the randomization check, no variable had to be included as covariate in the binomial logistic regression analyses because randomization seems successful.

The correlation analysis (see Table 2) showed that there was no significant correlation between the condition and healthy food choice (*r* = 0.04, *p* = 0.306) and between health consciousness and healthy food choice (*r* = 0.04, *p* = 0.309). This means that no significant relationship existed between the condition and healthy food choice and between health consciousness and healthy food choice. Also age (*r* = 0.08, *p* = 0.182), attitude toward the brand (*r* = −0.01, *p* = 0.466), and product familiarity (*r* = −0.08, *p* = 0.172) did not correlate significantly with healthy food choice. Nevertheless, visit frequency (*r* = −0.19, *p* = 0.012) and gender (*r* = 0.32, *p* < 0.001) correlated significantly with healthy food choice. Therefore, these variables will be included as covariates in the binomial logistic regression analyses.

### Hypotheses Testing

This subsection shows the results of the binomial logistic regression analyses. Multiple logistic models were fitted to the data to test for differences in the probability of choosing a healthy food product for young adults who are exposed to a menu with subtle, explicit, or no health information. The no health information condition was used as a referent in the first contrast test (see Table 3), to enable comparison between the experimental conditions with and without health information in order to test hypothesis 1. The subtle health information condition was used as a referent in the second contrast test (see Table 4), to enable comparison between the two experimental conditions with health information in order to test hypothesis 2.

The log of the odds of healthy food products being chosen by young adults was not significantly related to the health information condition (*p* = 0.064, see Table 3). When the no health information condition was taken as reference category, the results showed that there was no significant difference between the probability of choosing a healthy food product for young adults in the subtle health information condition and the probability in the no health information condition (*β* = −0.55, 95% CI [0.17, 1.98], Wald *χ*^2^ = 0.77, *p* = 0.381). In addition, the results showed that there was no significant difference between the probability of choosing a healthy food product for young adults in the explicit health information condition and the probability in the no health information condition (*β* = 0.75, 95% CI [0.79, 5.67], Wald *χ*^2^ = 2.19, *p* = 0.139).

When the subtle health information condition was taken as reference category, the results showed that there was a significant difference between the probability of choosing a healthy food product for young adults in the explicit health information condition and the probability in the subtle health information condition (*β* = 1.30, 95% CI [1.16, 11.57], Wald *χ*^2^ = 4.90, *p* = 0.027). The probability of choosing a healthy food product was 3.7 times higher for young adults in the explicit health information condition than for young adults in the subtle health information condition (see Table 4).

In the subtle health information condition, 11.9% of all young adults (5 out of 42) chose a healthy food product. In the explicit health information condition, 34.7% of all young adults (17 out of 49) chose a healthy food product. In the no health information condition, 17.6% of all young adults (9 out of 51) chose a healthy food product. Gender was the only variable that had a significant relation with healthy food choice (*β* = −2.88, *p* = 0.006). No moderation effects were found (*p* > 0.05) for gender and health consciousness, showing that the effects are similar among different groups.

To summarize, the results presented above show that hypothesis 1, which states that young adults are more likely to choose a healthy food product when they are exposed to a menu with health information about the healthy food product than when they are exposed to a menu without health information about the healthy food product, cannot be supported. In addition, hypothesis 2, which states that young adults are more likely to choose a healthy food product when they are exposed to a menu with explicit health information about the healthy food product than when they are exposed to a menu with subtle health information about the healthy food product, is supported.

## 4. Discussion

The main objective of the current study was to examine whether young adults who are exposed to a menu with subtle health information about healthy products would have a higher probability of choosing a healthy food product than young adults who are exposed to a menu with explicit health information or a menu without health information. First, health information was expected to have an effect on healthy food choice, with choosing a healthy food product being more likely for young adults who are exposed to a menu with health information about healthy products than for young adults who are exposed to a menu without health information. Second, the level of integration of health information into the menu context was expected to have an effect on healthy food choice, with choosing a healthy food product being more likely for young adults who are exposed to a menu with explicit health information about healthy products than for young adults who are exposed to a menu with subtle health information. Results confirmed this expectation, indicating that if health information is provided in an online restaurant setting, its level of integration into the menu context does matter. Next, moderation effects were analyzed for gender and health consciousness. However, as no such effects were found, it could be assumed that the effect of health information on healthy food choice was the same for the different groups.

These results suggest that it is possible that information alone may not have a direct effect on healthy food choice [8,9,54], but that the health information should be made explicit in a fast food environment in order to have an effect on the health perceptions of the healthy products, which in turn influences whether someone will choose one of these products [8]. The healthy food promotion model [8] describes that increased attention towards healthier foods through food promotion techniques increases the reinforcing value of healthier foods (e.g., liking and wanting), making them more likely to consume these foods. Health information could have been more effective for the salads when health information was also provided for the burgers to be more salient, to create a larger difference between the subtle health information provision and the information on the burgers [26]. Thus, providing health information for the healthy products alone was probably not enough to convince young adults not to choose an unhealthy product and to choose a healthy one instead. Therefore, for health information to be effective in inducing healthy food choice, it might be necessary to provide the information for both the healthy and unhealthy products.

These results provide fast food restaurants with a particular state-of-the-art tool to integrate promoting labels of healthy food products; make them more explicit. In general, food cues of unhealthy, palatable, and energy-dense foods in fast food restaurants, like images of these foods that are manipulated in order to look as palatable as possible, surrounded by bright colors and lights, induce attention of consumers to buy and consume these unhealthy products [6]. For example, what fast food restaurants could do is to provide specific nutritional labels or develop and present multiple healthier restaurant options in order to nudge consumers’ towards a healthier option. More specifically, it is suggested to provide explicit health information on the menu by mentioning the number of calories for the healthy menu products. With this information, young adults might not only be more willing to choose a healthy product, they also gain more knowledge about the healthiness of products.

The current study has several strengths. The first strength is that the order environment in which the manipulated menu of a popular fast food brand worldwide was presented, was similar to the environment of the menu that is displayed by the brand’s self-service system. This enhanced the ecological validity of the present study. The second strength is that the current study was able to distinguish between exposures to menus with subtle health information, explicit health information, and no health information in an experimental setting among a sample that was fairly representative of the Dutch population. The third strength is that, although it is not certain whether young adults would have actually consumed a certain product, healthy food choices were measured as opposed to attitudes toward, preferences for, or intentions to consume healthy food products. These strengths increased the internal and external validity of the study. In sum, this study aimed to include the methodological recommendations considered by Welch et al. [55] to improve our understanding of how healthy foods can be promoted more effectively.

Nonetheless, there are several limitations that might provide opportunities for future research. One limitation is that the participants were exposed to only four different food products. This was done to make sure that participants were able to easily distinguish the healthy from the unhealthy products and make a quick decision. However, in real life, young adults have access to more food types and not just salads and burgers. Therefore, the setting in the current study was not fully authentic to a truly naturalistic setting in which young adults choose food products in a menu in an actual restaurant. Nonetheless, the amount of people buying fast food online has increased immensely in the last decade, showing the importance of testing this new environment where food choices are being made. Next, it is not certain whether the findings are similar for other food types. Furthermore, the current study did not examine whether the perception of the tastiness of the healthy product(s) and the belief that tasty products are unhealthy mediate the effect of health information on healthy food choice.

## 5. Conclusions

Considering the importance for health and climate [1], there is an urgent need for research that focuses on the effectiveness of health information when provided for both the healthy and unhealthy food products [8,9], as information on healthy products may persuade people more likely to choose healthy food choices in a fast food environment. Future research should incorporate these factors in the assessment and analyze them when examining the effects of food labels on food choices. In addition, future research should also evaluate the impact of small changes to the forms of health information and the potential for generating an effect on healthy food choice. It might be possible that forms of health information that may be effective in promoting health will be identified. Lastly, future studies could dive deeper into gender differences regarding healthy food choices, or differences in health consciousness, as this might unravel bias for health information to a greater extent. Especially, different levels of health consciousness among people in lower socio-economic statuses might be apparent [8,54]. Promotion strategies tailored to a certain group of people might be effective in changing people’s consumption behavior.

## Figures and Tables

**Figure 1 ijerph-17-07139-f001:**
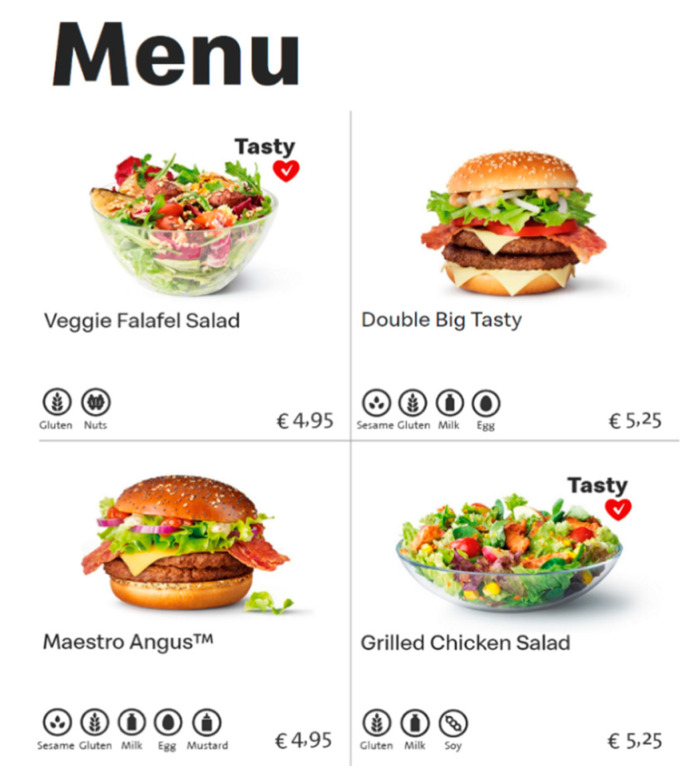
Condition 1: Subtle health information provided for a healthy menu product.

**Figure 2 ijerph-17-07139-f002:**
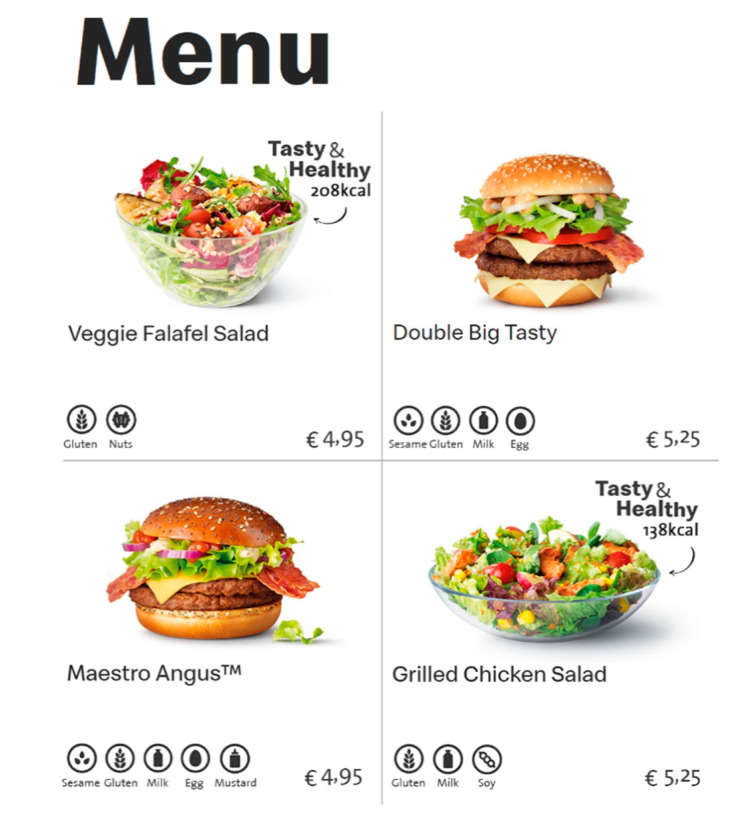
Condition 2: Explicit health information provided for a healthy menu product.

**Table 1 ijerph-17-07139-t001:** Variables measured for the subtle, explicit, and no health information conditions (*n* = 142).

Variable	Subtle Health (*n* = 42) Mean (SD)	Explicit Health (*n* = 49) Mean (SD)	No Health (*n* = 51) Mean (SD)	*p*-Value
Age	21.55 (1.73)	21.31 (1.64)	21.61 (1.94)	0.674
Attitude	4.50 (0.97)	4.51 (0.99)	4.54 (1.08)	0.977
Product Familiarity	3.83 (0.66)	3.71 (0.76)	3.90 (0.70)	0.415
Health Consciousness	5.06 (0.52)	5.10 (0.68)	5.19 (0.75)	0.622
Gender (women)	67%	78%	63%	0.261

Note. Continuous variables are presented by means and standard deviations (SD). Categorical variables are presented by percentages for the category with the highest percentage.

**Table 2 ijerph-17-07139-t002:** Correlation matrix of predictor and outcome variables.

	Condition	Age	Attitude	Product Familiarity	Health Consciousness	Gender
Condition						
Age	0.02					
Attitude	0.02	−0.03				
Product Familiarity	0.05	−0.06	**0.30**			
Health Consciousness	0.08	**0.19**	−0.09	0.06		
Gender	−0.04	−0.07	**−0.16**	**−0.13**	0.07	
Healthy Choice	0.04	0.08	−0.01	−0.08	0.04	**0.32**

Note. Significant (*p* < 0.05) correlations at the 1-tailed level are presented in **boldface.**

**Table 3 ijerph-17-07139-t003:** Results of the logistic regression models for the logit to choose a healthy food product with the control condition as reference category (*n* = 142).

	Model 1	
Variable	OR	SE
Constant	0.26	1.90
Condition (ref: control)		
Condition (subtle health)	0.58	0.63
Condition (explicit health)	2.11	0.50
Health Consciousness	1.07	0.35
Gender (ref: women)	0.06 *	1.04
Nagelkerke pseudo *R*^2^	0.25	

Note. OR indicates odds ratio. SE indicates standard error. * Indicates significance (*p* < 0.01).

**Table 4 ijerph-17-07139-t004:** Results of the logistic regression models for the logit to choose a healthy food product with the subtle health condition as referent (*n* = *142*).

	Model 2	
Variable	OR	SE
Constant	0.15	1.84
Condition (ref: subtle health)		
Condition (explicit health)	3.67 *	0.59
Condition (no health)	1.74	0.63
Health Consciousness	1.07	0.35
Gender (ref: women)	0.06 **	1.04
Nagelkerke pseudo *R*^2^	0.25	

Note. OR indicates odds ratio. SE indicates standard error. * Indicates significance (*p* < 0.05). ** Indicates significance (*p* < 0.01).

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
