# Peer review of "Taste and Health Information on Fast Food Menus to Encourage Young Adults to Choose Healthy Food Products: An Experimental Study"

_ijerph, 2020, doi:10.3390/ijerph17197139_

Round 1

Reviewer 1 Report

General Comments and suggestions for the authors

It is a manuscript that explores the effect of the presence of health information and its integration into a fast-food menu context on 16 young adults´ healthy food choices. It provides insights into healthy food choices. Although the proposal is not a clinical trial, the design incorporates intervention assessment. It could be useful to take a closer look at the guidance material about this study design, as indicated below. I recommend that the authors point out some issues:  

  • Information about authors conflict of interest;
  • It was not shown details about the allocation ratio. Authors could also give information about eligibility criteria and reasons.
  • Completely defined pre-specified primary and secondary outcome measures, including how and when they were assessed
  • Blinding procedures should be addressed expressing who generated the random allocation sequence, who enrolled participants, and who assigned participants to interventions.
  • Results could be presented as recommended by Bibro et al. (2020) and Welch et al. (2013).

Suggested references:

Bilbro NA, Hirst A, Paez A, Vasey B, Pufulete M, Sedrakyan A, McCulloch P; IDEAL Collaboration Reporting Guidelines Working Group. The IDEAL Reporting Guidelines: A Delphi Consensus Statement Stage-specific recommendations for reporting the evaluation of surgical innovation. Ann Surg. 2020.

Welch RW, Antoine JM, Berta JL, Bub A, de Vries J, Guarner F, Hasselwander O, Hendriks H, Jäkel M, Koletzko BV, Patterson CC, Richelle M, Skarp M, Theis S, Vidry S, Woodside JV; International Life Sciences Institute Europe Functional Foods Task Force. Guidelines for the design, conduct and reporting of human intervention studies to evaluate the health benefits of foods. Br J Nutr. 2011;106 Suppl 2:S3-15

Author Response

General Comments and suggestions for the authors

C1: It is a manuscript that explores the effect of the presence of health information and its integration into a fast-food menu context on 16 young adults´ healthy food choices. It provides insights into healthy food choices. Although the proposal is not a clinical trial, the design incorporates intervention assessment. It could be useful to take a closer look at the guidance material about this study design, as indicated below. I recommend that the authors point out some issues:  

  • Information about authors conflict of interest;
  • It was not shown details about the allocation ratio. Authors could also give information about eligibility criteria and reasons.
  • Completely defined pre-specified primary and secondary outcome measures, including how and when they were assessed
  • Blinding procedures should be addressed expressing who generated the random allocation sequence, who enrolled participants, and who assigned participants to interventions.
  • Results could be presented as recommended by Bibro et al. (2020) and Welch et al. (2013).

A1: We agree with the reviewer that this information could be more clear although the study was no clinical trial, therefore we have revised this information (if not already included in the manuscript) in the methodology section mostly according the provided suggestions. We thank the reviewer also for the suggested research papers, in particular the 2nd one was relevant and we have included this in our manuscript.

Reviewer 2 Report

It is an interesting work on an extremely topical topic.

The reading is very difficult because the same concepts are repeated many times both in the introduction and in the discussion.

Introduction: the authors believe that the food industry needs to change and stimulate the consumer to take healthier foods. It is clear that this is impossible because the modern consumer has to deal with three forces that seemingly struggle with each other but in reality have the sole purpose of exploiting citizens to ensure maximal profits. The first force is the food industries, whose simple objective is to make the subject eat as much as possible. The second force is the dieting industry, now called the health industry, which should have the objective of improving well-being and preventing morbidity but actually disorients and exploits subject to maximise profits. The third force is the pharmaceutical industry, to which we can add the even more harmful food supplement industry. These three forces act synergistically and become increasingly stronger, influencing government health guidelines and policies, as described in M. Nestle's 2013 book. So the authors' aim should be to suggest to government authorities a state-of-the-art tool to encourage consumers to buy healthier food in fast-food restaurants.

Methods: The difference between the proposed images is very limited. Basically there is only the number of calories in the two healthiest food choices. Why was it not chosen to include more information such as the amount of salt or saturated fat? The work would have been much more interesting if done in fast-food and not through the online questionnaire. 

Statistics: It can be difficult for a non-worker to understand the various "steps". The explanation should be simplified.

Discussion: As said, it is written in an unclear and not very synthetic way. The concepts are repeated over and over again and it is difficult to understand what the practical and developmental implications of other scientific work are. 

Author Response

C1: It is an interesting work on an extremely topical topic.

A1: We thank the reviewer for this positive comment and we agree that more research needs to be conducted in this area.

C2: The reading is very difficult because the same concepts are repeated many times both in the introduction and in the discussion.

A2: We have revised the manuscript to make it easier to read, both in the introduction and in the discussion.

C3: Introduction: the authors believe that the food industry needs to change and stimulate the consumer to take healthier foods. It is clear that this is impossible because the modern consumer has to deal with three forces that seemingly struggle with each other but in reality have the sole purpose of exploiting citizens to ensure maximal profits. The first force is the food industries, whose simple objective is to make the subject eat as much as possible. The second force is the dieting industry, now called the health industry, which should have the objective of improving well-being and preventing morbidity but actually disorients and exploits subject to maximise profits. The third force is the pharmaceutical industry, to which we can add the even more harmful food supplement industry. These three forces act synergistically and become increasingly stronger, influencing government health guidelines and policies, as described in M. Nestle's 2013 book. So the authors' aim should be to suggest to government authorities a state-of-the-art tool to encourage consumers to buy healthier food in fast-food restaurants.

A3: We are well aware of Marion Nestle’s great work and we have revised the introduction according the suggestion provided by the reviewer. We think the main aim as has been suggested by the reviewer is very useful and we have integrated this at the end of the introduction and also in the discussion.

C4: Methods: The difference between the proposed images is very limited. Basically there is only the number of calories in the two healthiest food choices. Why was it not chosen to include more information such as the amount of salt or saturated fat? The work would have been much more interesting if done in fast-food and not through the online questionnaire. 

A4: Indeed the differences between the images are very limited for specific reasons. We have decided to use only very subtle changes on the images to increase the probability that it can be used by the industry eventually. As the reviewer has stated in C3, the main aim of this study should be a state-of-the-art tool to direct consumers’ decisions towards a healthier option in fast-food restaurants; if minor changes in the information provision could have this effect, this will help the restaurants with easily implementable options. In addition, we agree that the work would have been more interesting if we would have conducted the study in an actual fast-food, but this would lead to practical difficulties that were not solvable in this study (i.e., implementation in real restaurants, approval owner of the restaurants). We also included this in the discussion as one of the limitations. Nonetheless, currently, in particular during the COVID-19 crisis, we have seen that an increasing number of people buy online fast-food for delivery, which is a similar situation as we have created. Next, the interface of fast-food restaurant system that we have used was similar to that of a drive-in and home-delivery system thereby increasing ecological validity. Therefore, we strongly believe the current study has important merits to scientific research. 

C5: Statistics: It can be difficult for a non-worker to understand the various "steps". The explanation should be simplified.

A5: We have revised the statistical information in order to be more explanatory and direct.

C6: Discussion: As said, it is written in an unclear and not very synthetic way. The concepts are repeated over and over again and it is difficult to understand what the practical and developmental implications of other scientific work are. 

A6: We thank the reviewer for this suggestion and we have deleted the repetition of concepts and included the practical and developmental implications of this work.

Round 2

Reviewer 1 Report

The revised material is better and included recommended topics. Nevertheless, it would be nice to reconsider the conclusion to show the possible influence of the information on healthy products and the consumer's decision. So, the sentencing could be rephrased as "information on healthy products may turn people more likely to choose healthy food choices in a fast-food environment."

Author Response

C1: The revised material is better and included recommended topics. Nevertheless, it would be nice to reconsider the conclusion to show the possible influence of the information on healthy products and the consumer's decision. So, the sentencing could be rephrased as "information on healthy products may turn people more likely to choose healthy food choices in a fast-food environment."

A1: We thank the reviewer for this suggestion, also in line with the suggestions of reviewer 2. We have revised the discussion and conclusion accordingly, in particular rephrasing the concluding sentence, but also more general by integrating the influence of consumers´ decisions in a fast-environment. 

Reviewer 2 Report

The authors have definitely improved the paper following my considerations.

The discussion could go into more detail on how the food industry influences the unhealthy nutritional choices of young consumers.

Some suggestions could be added on how to break this unhealthy influence perhaps through the introduction of particular nutritional labels or restaurant options.

Author Response

C1: The authors have definitely improved the paper following my considerations.

A1: We thank the reviewer for this positive comment. And for the improvements the reviewer suggested. We strongly believe the manuscript has improved significantly due to the responses of the Reviewers.

C2: The discussion could go into more detail on how the food industry influences the unhealthy nutritional choices of young consumers.

C3: Some suggestions could be added on how to break this unhealthy influence perhaps through the introduction of particular nutritional labels or restaurant options.

A2&A3: We have done this accordingly, also following the comment by Reviewer 1.